# Effect of Different Drying Methods on the Quality of *Oudemansiella raphanipes*

**DOI:** 10.3390/foods13071087

**Published:** 2024-04-01

**Authors:** Shuting Hou, Defang Zhang, Dongmei Yu, Hao Li, Yaping Xu, Wuxia Wang, Ruiting Li, Cuiping Feng, Junlong Meng, Lijing Xu, Yanfen Cheng, Mingchang Chang, Xueran Geng

**Affiliations:** 1College of Food Science and Engineering, Shanxi Agricultural University, Jinzhong 030801, China; houshuting0829@163.com (S.H.); zdf18434003425@163.com (D.Z.); sxndydm2084@163.com (D.Y.); lihao980423@163.com (H.L.); yc_xuyaping@163.com (Y.X.); wwx17303451304@163.com (W.W.); lrtt0625@163.com (R.L.); ndfcp@163.com (C.F.); mengjunlongseth@126.com (J.M.); xulijing383942909@163.com (L.X.); cyf2341986@163.com (Y.C.); gengxueran2007@163.com (X.G.); 2Shanxi Key Laboratory of Edible Fungi for Loess Plateau, Jinzhong 030801, China; 3Shanxi Edible Fungi Engineering Technology Research Center, Jinzhong 030801, China

**Keywords:** *Oudemansiella raphanipes*, drying, HS-SPME-GC-MS, volatile compounds, polysaccharide, physical and chemical properties

## Abstract

In this study, we used fresh *Oudemansiella raphanipes* as raw materials and pre-treated through hot air drying (HD), infrared radiation drying (ID), and vacuum freeze drying (VD) to investigate the effects of different drying methods on the rehydration rate, appearance quality, microstructure, and volatile flavor components of the dried products, as well as to determine the physicochemical properties and bioactivities of the polysaccharides in the dried *O. raphanipes.* The results showed that the VD *O. raphanipes* had the highest rehydration rate and the least shrinkage in appearance, and it better maintained the original color of the gills, but their aroma was not as strong as that of the HD samples. The scanning electron microscopy results indicate that VD maintains a good porous structure in the tissue, while HD and ID exhibit varying degrees of shrinkage and collapse. Seventy-five common volatile substances were detected in the three dried samples, mainly alkanes, alcohols, and esters. The polysaccharides (PS-H, PS-I, and PS-V) extracted from the dried samples of these three species of *O. raphanipes* had similar infrared spectral features, indicating that their structures are basically consistent. The highest yield was obtained for PS-V, and the polysaccharide content and glucuronic acid content of PS-I were higher than those of the remaining two polysaccharides. In addition, PS-V also showed better antioxidant activity and inhibitory activity against α-glucosidase as well as α-amylase. In conclusion, among the above three drying methods, the quality of *O. raphanipes* obtained by vacuum freeze drying is the best, and this experiment provides a theoretical basis for the selection of drying methods for *O. raphanipes*.

## 1. Introduction

The nutrient content and appearance quality of food are easily affected by processing technology, and drying, as an efficient, safe, and inexpensive processing method, has been widely researched and applied [1]. Drying is mainly through dehydration to reduce the water content and microbial growth in fresh fruits, vegetables, and mushrooms to achieve the purpose of prolonging the storage time of food and improving the value of goods [2]. In recent years, China’s edible mushrooms’ planting scale has continued to expand, and the number of varieties continues to increase. At present, China produces about 967 species of edible mushrooms and is the world’s major edible mushrooms producer, accounting for almost 50% of the world’s cultivable edible mushrooms species [3]. Edible mushrooms will have a high water content, after harvesting, if improper storage is very susceptible to corruption and deterioration. In addition, the storage and transportation of fresh edible mushrooms are a cumbersome and costly process prone to mechanical damage, thus severely limiting their distribution in the marketplace [4]. Drying can reduce the water content and water activity of edible mushrooms to a certain degree, effectively solving the problem of short shelf life and reducing the cost of storage and transportation; edible mushrooms in the drying process can generate new volatile substances to improve their characteristic flavor [5]. Common drying techniques for edible mushrooms include hot air drying, infrared radiation drying, and vacuum freeze drying. Different drying methods had a significant effect on the flavor and nutritional value of the resulting products [6]. Research has shown that there are significant differences in the nutritional content of *O. raphanipes* after hot air drying, heat pump drying, and vacuum freeze drying. Specifically, the protein and fat content of *O. raphanipes* are highest after heat pump drying, at 26.07% and 1.37%, respectively. For carbohydrates, the highest content is 32.57% after vacuum freeze drying. The dietary fiber content of *O. raphanipes* can reach 33.11% after heat pump drying, significantly higher than the other two dried samples [7]. Similarly, different drying methods can also significantly affect the flavor of edible mushrooms. LI weilan et al. [8] used hot air drying, vacuum freeze drying, and vacuum drying to process *Boletus*. They found that hot air drying can promote the accumulation of umami substances, vacuum freeze drying can retain the flavor and umami substances of *Boletus* more completely, and vacuum drying can produce and accumulate more flavor substances to a greater extent. Therefore, the choice of drying method is also very important. Hot air drying (HD) is simple and inexpensive, and at the same time facilitates the formation of the characteristic flavor substances in *Lentinula Edodes* due to enzymatic and Maillard reactions; however, it also leads to a partial loss of nutrients and the appearance of the final product [9]. Compared with hot air drying, Xie et al. [10] used infrared radiation drying (ID) technology to dry *Lentinula Edodes* and found that it not only improved the drying speed but also maintained the quality of *Lentinula Edodes*. Studies have shown that the high-quality characteristics of products after vacuum freeze drying (VD) are attributed to the low temperature and moisture transfer modes [11]. Therefore, it is widely used to obtain high-quality dried mushrooms, including *Lentinula Edodes*, *Agaricus bisporus*, and *Pleurotus eryngii* [12].

*Oudemansiella raphanipes* is also known as *Oudemansiella radicata*, *Hymenopellis* [13]. *O. raphanipes* is widely distributed in regions with high temperatures, such as Southeast Asia and Africa, and in China, it is mainly distributed in Yunnan, Guangxi, Sichuan, Guangdong, Fujian, and other provinces. Since the first report on the cultivation of *O. raphanipes* in 1966, with the in-depth study of its biological characteristics and the improvement in cultivation technology, gradually realized the indoor cultivation, greenhouse cultivation, and factory cultivation without mulching and other cultivation methods [14]. *O. raphanipes* is suitable for growth in neutral or slightly acidic environments, with the optimal pH for mycelial growth being 6.0~7.5 [15]. In recent years, due to people’s attention and preference, *O. raphanipes* has been cultivated and produced in large quantities [16]. Ou Shengping et al. [17] conducted a quantitative study on the nutrients contained in *O. raphanipes*, and the data showed that *O. raphanipes* contains a large amount of proteins and carbohydrates, which accounted for 32.12% and 24.25% of their total content, respectively, which directly indicates that *O. raphanipes* is highly nutritious. Research by Du Na et al. [18] has shown that *O. raphanipes* also contains rich mineral elements, such as phosphorus, potassium, calcium, magnesium, iron, and zinc. This indicates that *O. raphanipes* is a high-protein and nutritionally rich edible fungus. It has a high content of umami amino acids and can be developed into a raw material for natural healthy seasonings. LU Caihui et al. [19] reported that *O. raphanipes* contains a variety of antioxidant bioactive substances, such as polysaccharides, brass, and polyphenols. By measuring the antioxidant activity of the aqueous and alcoholic extracts of *O. raphanipes*, it was confirmed that both extracts have good scavenging abilities for DPPH radicals, hydroxyl radicals, and ABTS radicals. Among them, polysaccharides, as one of the most important macromolecules in organisms, are able to participate in cell growth and senescence and have a wide range of pharmacological activities [20]. Studies have shown that polysaccharides are the main antioxidant active components in *O. raphanipes* [21], as well as having various functions, such as hypolipidemic, enhancement in body immunity, antibacterial and anti-inflammatory, antitumor, and protection of the liver [22]. Therefore, to better preserve the flavor and nutrients of *O. raphanipes* and to reduce losses during storage and transportation, drying techniques can be used.

Currently, studies on the drying of edible mushrooms mainly focus on the species of *Lentinula Edodes*, *Pleurotus eryngii*, and *Agrocybe aegerita*, while there are fewer studies related to the drying of *O. raphanipes*, especially on the effects of different drying methods on *O. raphanipes*. In view of this, this experiment investigated the effects of hot air drying, infrared radiation drying, and vacuum freeze drying on the quality of *O. raphanipes* from three aspects: (1) evaluation of its rehydration rate, appearance quality, and microstructure; (2) HS-SPME-GC-MS was used to analyze the effects of different drying methods on the volatile flavor components of *O. raphanipes*; (3) the effect of different drying methods was further investigated by determining the physicochemical properties of polysaccharides. In order to optimize the drying method that can maximally maintain the excellent quality of *O. raphanipes* and provide a theoretical basis for the selection of the drying processing method of *O. raphanipes*, this study will lay the foundations for its further deep processing.

## 2. Materials and Methods

### 2.1. Materials and Reagents

Fungal material: Fresh *O. raphanipes* was purchased from Yunnan Xingmin Agricultural Technology Co., Ltd. (Kunming, Yunnan, China).

Material preparation: Select a number of mature, mold- and pest-free infestation, complete skin, consistent appearance (including color and size) of *O. raphanipes*, dispense for subsequent experiments.

Reagents: 1,1-diphenyl-2-picrylhydrazyl (DPPH) and acarbose (Ac) were purchased from Sigma-Aldrich (St Louis, MO, USA). Vitamin C (Vc), α-amylase, α-glucosidase (10 U/mg) were obtained from Yuan Ye (Shanghai, China). All other chemicals used in this study were of analytical reagent grade from Tianjin Kaitong Chemical Reagent Factory (Tianjin, China).

Instruments: GZX-9240MBE Electric Blast Drying Oven (Shanghai, China; Shanghai Boxun Industry Co., Ltd.); Alpha 2–4 Freeze Dryer (Osterode am Harz, Lower Saxony, Germany; Martin Christ Company); YHG 300-S Far Infrared Drying Oven (Shanghai, China; Shanghai Precision Instruments Co., Ltd.); CM-600D Colorimeter (Nakano, Tokyo, Japan; Konica Minolta Company); JSW-6490LV Scanning Electron Microscope (Musashino, Japan; JEOL Company); WFM-10 Ultrafine Crushing Oscillating Grinder (Jingzhou, Hubei, China; Xiangda Machinery Manufacturing Co., Ltd.); MultifugeX1R High-Speed Centrifuge (Waltham, MA, USA; THERMO Limited Company); RV10 Rotary Evaporator (Staufen im Breisgau, Germany; IKA Company); FT-IR 650 Fourier-Transform Infrared Spectrometer (Waltham, MA, USA; Thermo Fisher Company); UV9100 A Ultraviolet-Visible Spectrophotometer (Beijing, China; LabTech Instruments Co., Ltd.).

### 2.2. Drying Methods

There are three different drying methods, namely hot air drying (HD), infrared radiation drying (ID), and vacuum freeze drying (VD). In the preparation of the above materials, three 50 g samples of *O. raphanipes* were taken as experimental samples. Table 1 shows the parameter settings of each drying method. Specific operations are as follows.

During hot air drying (HD), 50 g *O. raphanipes* was laid flat in the oven. In order to dry them fully, the samples were placed in a single layer without overlapping. The temperature was set at 50 °C, and the convection rate was 0.3 m^3^/s. At the beginning of drying, samples were taken at intervals of 2 h to determine the moisture content. When the dry base moisture content of the sample was less than 13%, the drying was ended.

During infrared radiation drying (ID), weigh 50 g of *O. raphanipes* and lay it on the drying screen in the form of a single layer. Select the far infrared radiation mode, with radiation intensity of 2 W/cm^2^, temperature set at 50 °C, radiation distance between the far-infrared radiation plate and the material tray of 10 cm, and the wind speed above the material at 1.5 m/s. Samples were taken at an interval of 2 h to determine the dry base water content of the sample. When it was less than 13% of the safe water content, the drying was finished.

When vacuum freeze drying (VD) was performed, 50 g of *O. raphanipes* was also weighed and placed in a low-temperature freezer (−20 °C) for quick freezing for 6 h and then frozen in an environment of −80 °C for 12 h. The sample was placed on the material shelf of a vacuum freeze-drying machine (cold trap temperature of −50 °C and vacuum degree of 1.5 hPa). Samples were taken at intervals of 2 h to determine the moisture content. When the dry base moisture content of the sample was less than 13%, the drying was finished.

Dry moisture content of *O. raphanipes*: dry moisture content W = (m_1_ − m_2_)/m_2_ × 100%, where m_1_ represents the starting mass of fresh material and m_2_ represents the mass of dried material.

In order to reduce the error of the subsequent test, the final water content of the three dry samples is controlled at the same level below 13%.

### 2.3. Determination of Drying Characteristic Indicators

#### 2.3.1. Appearance Quality Evaluation

Appearance quality was evaluated with reference to the method proposed by Fu Qingquan et al. [23], in which ten trained appearance quality assessors were invited to evaluate the samples of *O. raphanipes* after drying by HD, ID, and VD from the three aspects of gill color, aroma, and shrinkage, using a weighted method. The total score for this indicator of appearance quality evaluation is 40 points, and the specific scoring criteria are shown in Table 2.

Color difference analysis uses the *CM-600D* colorimeter to measure the color difference of *O. raphanipes* treated with three different drying methods. The instrument is color-corrected with a white board, and the reflection mode is selected to detect the same position in the three samples. First, the color of the sample before drying is measured as the baseline measurement result in order to calculate the color difference later. After the sample is dried, the measurement window of the colorimeter is placed at the same position for measurement, and three repeated tests are carried out to obtain the *L**, *a**, *b** values and record them. The color difference of each sample is calculated based on the baseline measurement results.

#### 2.3.2. Rehydration Rate

Referring to the method in [24] and making certain modifications, take a beaker, add 200 mL of deionized water, and then add about 3 g of *O. raphanipes* samples dried by HD, ID, and VD, respectively. Rehydrate at a constant temperature of 60 °C, and remove the samples at intervals of 20 min. Place the sample on a hollow mesh and drain for 3 min, allowing the surface moisture to completely evaporate. Weigh it and repeat this process 6 times for a total of 120 min in water bath. The formula for the rate of rehydration (R) is as follows:R (%) = G_2_/G_1_ × 100

G_1_ is the initial mass of dried *O. raphanipes*; G_2_ is the mass of rehydrated *O. raphanipes.*

#### 2.3.3. Microstructure

Take *O. raphanipes* dried with HD, ID, and VD, and grind them into fine powder. The samples are magnified 1500- and 4000-times, respectively, and images captured to compare the differences in their microstructures [25].

### 2.4. HS-SPME-GC-MS Analysis

Sample pretreatment: Weigh about 1 g of the sample and transfer it to a 20 mL headspace injection bottle for HS-SPME-GC-MS analysis.

HS-SPME extraction conditions: Under constant temperature conditions of 60 °C, shake for 15 min at a speed of 450 rpm (5 s on, 2 s off). A 50/30 μm DVB/CAR on PDMS extraction tip was inserted into the headspace portion of the sample, headspace extracted for 40 min, desorbed at 230 °C for 5 min, and then separated and identified by GC-MS.

Chromatographic conditions: HP-5MS Capillary Columns (30 m × 0.25 mm × 0.25 μm, Agilent J&W Scientific, Folsom, CA, USA), the carrier gas is high-purity helium gas (the carrier gas is high-purity helium gas), constant flow rate of 1.0 mL/min, the inlet temperature is 230 °C, there is no split injection, the solvent is delayed for 1.5 min. Program heating: hold at 40 °C for 1 min, raise to 210 °C at 5 °C/min, and hold for 5 min.

Mass Spectrometry Conditions: For the electron bombardment ion source (EI), the ion source temperature is 230 °C, the fourth pole temperature is 150 °C, and the electron energy is 70 eV. Scanning mode is full-scan mode (SCAN); quality scanning range *m*/*z* 20–650.

Qualitative and quantitative analysis: For headspace injection GC-MS experiments to study volatile substances, we used the NIST database for substance characterization. For quantitative analysis, the total peak area was normalized for all peak signal intensities (peak areas) in each sample (the signal intensity of each peak was used to convert to a relative intensity in this spectrogram and, after normalizing the data, multiplied by 10,000). After data normalization, redundancy reduction and peak merging were performed to obtain the data matrix.

### 2.5. Extraction of Polysaccharides

Referring to the method of Guo et al. [26] with slight modifications, the dried powders of three types of *O. raphanipes* are added to deionized water at a material-to-liquid ratio of 1:30 (*W*/*V*), mixed and then heated for 2 h and 30 min. After cooling, the extract is filtered twice, and the filtrate is collected and concentrated using a rotary evaporator. A prepared solution of potassium ferrocyanide and zinc acetate is added to remove proteins. After centrifugation, the supernatant is taken for dialysis. Anhydrous ethanol is added and left to stand for 12 h, and then the mixture is centrifuged to collect the precipitate. The precipitate is redissolved in an appropriate amount of distilled water and then freeze dried to obtain polysaccharide samples of the three types of *O. raphanipes.* They are, respectively, named hot-air-dried *O. raphanipes* polysaccharide (PS-H), infrared radiation dried *O. raphanipes* polysaccharide (PS-I), and vacuum freeze-dried *O. raphanipes* polysaccharide (PS-V).

### 2.6. Structural Properties of Polysaccharides

#### 2.6.1. Fourier Infrared and Ultraviolet Spectral Scanning of Polysaccharides

According to a previous study [27], samples of three polysaccharides to be tested were prepared using the potassium bromide compression method. Fourier-transform infrared (FTIR) spectra were scanned in the wave number range of 4000 to 500 cm^−1^ to compare the peak spectra of these three samples. Prepare 1 mg·mL^−1^ aqueous solutions of PS-H, PS-I, and PS-V, respectively, and use a UV-visible spectrophotometer to detect the UV-visible absorption spectra of the samples in the range of 190–800 nm [28].

#### 2.6.2. Electron Microscopy (SEM)

Refer to the method in Section 2.3.3 above for detection.

### 2.7. Determination of the Chemical Composition of Polysaccharides

Formula: polysaccharide yield (%) = weight of polysaccharide after freeze drying (g)/total weight of *O. raphanipes* seed solid powder (g) × 100%. The total sugar content in polysaccharides was determined by the phenol–sulfuric acid method [29], and the standard curve was plotted with different concentrations of glucose standards. Determination of protein content in polysaccharides was carried out by the method of Caulmers Brilliant Blue staining [30], and standard curves were plotted with different concentrations of bovine serum standard solution. Glycuronic acid content was determined by the sulfuric acid–carbazole method [31], and the standard curve was plotted with different concentrations of galacturonic acid standard solutions. The polyphenol content was determined by the forintol method [32], and the standard curve was plotted with different concentrations of gallic acid standards.

### 2.8. Antioxidant Capacity of Polysaccharides

#### 2.8.1. DPPH Free-Radical Scavenging Capacity

The three polysaccharides, PS-H, PS-I, and PS-V, were prepared as aqueous solutions of polysaccharides with concentration gradients of 0, 0.25, 0.5, 1, 2, and 4 mg·mL^−1^, respectively. Pipette 2 mL of polysaccharide solution in a 10 mL EP (Eppendorf) tube, add 2 mL of 0.2 mM DPPH-methanol solution, mix well, and react for 30 min, avoiding light, and then measure the absorbance value at 517 nm and use VC (vitamin C) as positive control [33]. The formula is as follows:$$\mathrm{DPPH}\;\mathrm{radical}\;\mathrm{scavenging}\;\mathrm{rate}\;(\%)=\left(1-\frac{A_{1}-A_{2}}{A_{0}}\right)\times 100$$

*A*_0_: absorbance value of 2 mL DPPH-methanol solution and 2 mL methanol solution; *A*_1_: absorbance values of 2 mL polysaccharide solution and 2 mL methanol solution; *A*_2_: absorbance values of 2 mL DPPH-methanol solution and 2 mL polysaccharide solution.

#### 2.8.2. •OH Radical Scavenging Capacity

The configured polysaccharide solution was placed in a 10 mL EP tube as above. Add 2 mL of 9 mmol/L FeSO_4_ solution, 2 mL of 9 mmol/L salicylic acid–ethanol solution, mix well, and let it stand at room temperature for 10 min. We added 2 mL of 9 mmol/L H_2_O_2_ solution, mixed well, and let it stand at room temperature for 30 min, and then the absorbance value was measured at 510 nm, using VC (vitamin C) as the positive control and double-distilled water as the blank control [34]. The formula is as follows:$$•\mathrm{OH}\;\mathrm{radical}\;\mathrm{scavenging}\;\mathrm{rate}\;(\%)=\left(1-\frac{A_{1}-A_{2}}{A_{0}}\right)\times 100$$

*A*_0_: The absorbance value of the solution without sample addition; *A*_1_: absorbance value after reaction with sample solution added; *A*_2_: absorbance value without adding salicylic acid ethanol solution.

### 2.9. Determination of the Inhibitory Effect of Polysaccharides on α-Glucosidase as Well as α-Amylase

According to the method of [35], the three polysaccharides PS-H, PS-I, and PS-V were prepared as aqueous solutions of the polysaccharides with concentration gradients of 0, 0.25, 0.5, 1, 2, 4, and 8 mg·mL^−1^, respectively. Then, 1 mL of polysaccharide solution was pipetted into a 5 mL EP tube, 500 μL of 1 U·mL^−1^ glucosidase was added, mixed well, and incubated at 37 °C for 10 min, followed by the addition of 1 mL of 5 mmol/L PNPG and incubation at 37 °C for 30 min, and the reaction was terminated by the addition of 2 mL of 0.2 mol/L Na_2_CO_3_. Absorbance values were measured at 405 nm and acarbose was used as a positive control. The inhibition of α-amylase activity by polysaccharides was determined according to the method in [36]. The three polysaccharides PS-H, PS-I, and PS-V were prepared as aqueous solutions of the polysaccharides with concentration gradients of 0, 0.25, 0.5, 1, 2, 4, and 8 mg/mL, respectively. Transfer 1.0 mL of polysaccharide solution into a 10 mL EP tube, add 0.5 mL of 5 U/mL amylase solution, incubate at 37 °C for 10 min, then add 0.50 mL of 0.5% starch solution, incubate at 37 °C for 10 min, and finally add 4 mL of DNS boiling water bath for 5 min. After cooling, the absorbance value was detected at 540 nm, with AC (acarbose) as the positive control.

### 2.10. Data Analysis

All experiments were repeated at least 3 times, and the results were taken as mean ± standard deviation. IBM SPSS Statistics 26.0 software was used to process the data, and the difference was analyzed for significance by ANOVA, where *p* < 0.05 was significant. We used Origin 2021 software to draw charts.

## 3. Results and Analysis

### 3.1. Effects of Different Drying Methods on the Appearance Quality of O. raphanipes

With reference to Section 2.3.1, the appearance quality of *O. raphanipes* treated with different drying methods was evaluated in terms of gill color, aroma, and degree of shrinkage, and the scoring results are shown in Table 3.

As can be seen from Table 3, VD scored the highest in terms of gill color, which allows the samples to be kept under vacuum and non-high-temperature conditions, reducing the possibility of enzymatic as well as non-enzymatic browning of *O. raphanipes* and, therefore, maintaining the original color of the gills to the maximum extent possible. On the contrary, HD drying temperatures were high and prolonged, and the samples underwent severe browning, hence the darkest color of the gill. The effect of ID on the color of the gills is in between, probably due to the low energy of infrared photons, which can keep the components of the raw material stable during the heating process, so the degree of influence on its chemical properties is small. Wang et al. [37] performed a similar drying treatment on *F. velutipes* and showed that the VD treatment had the least effect on the color change in *F. velutipes.*

*O. raphanipes* treated with HD had a more intense flavor compared to ID and VD. This is similar to the findings of Liu et al. [38] that HD creates a high-temperature environmental condition under which chemical components such as fatty acids in *O. raphanipes* may undergo decomposition reactions to produce volatiles with distinctive aromas, and, thus, HD may increase the aroma of dried products. In a certain sense, ID can also increase the amount of volatiles, and VD does not adversely affect the aroma after drying.

After HD and ID drying, the surface of *O. raphanipes* showed an obvious wrinkling phenomenon; the stipe was thinned, and the cap was narrowed. And after drying by VD, its volume did not change significantly from fresh, so it scored the highest for this indicator of shrinkage. The degree of wrinkling is closely related to the internal structure of the *O. raphanipes* VD drying. The moisture in the fruiting body forms ice crystals at low temperatures, sublimates into water vapor, and forms a microporous structure, so it has little effect on the appearance of the mushroom body. HD and ID samples, under prolonged heating and drying conditions, destroy the original spatial structure, shrink in volume, harden on the surface, and crumple severely.

Based on the above analysis, among these three drying methods, vacuum freeze drying scored the highest in terms of appearance quality, and this method maximizes the preservation of the original color, aroma, and appearance of the gills of the *O. raphanipes.*

Hot air drying, infrared drying, and vacuum freeze drying were used to dry to constant weight (±0.02 g), after 10 h, 10 h, and 24 h, respectively, until the moisture content of the sample was reduced to about 6%. According to Table 4, it can be seen that the rehydration rate of *O. raphanipes* dried by vacuum freeze drying is the highest, and the rehydration rate of dried products dried by infrared radiation is slightly higher than that of hot air drying. The high rehydration ability of VD dried products may be related to the principle of freeze drying. Vacuum freeze drying can directly sublimate the water in the fruiting body, maintaining the original voids inside the *O. raphanipes.* Therefore, the dried product has a porous structure, which helps to absorb water. Therefore, this drying method has a high rehydration rate. Both HD and ID are dried by heating, and the prolonged high temperature deforms the cells of the fruiting body, damages the tissue structure, and hardens the surface crusts, which is not conducive to rehydration.

As shown in Table 4, the *L** values of hot air drying, infrared radiation drying, and vacuum freeze drying are 66.68, 65.17, and 79.26, respectively. The *L** value of *O. raphanipes* before drying is 93.11. This indicates that all three drying techniques resulted in a decrease in the brightness of *O. raphanipes*. The increase in *a** and *b** values is due to the formation of brown polymers in the relevant components of *O. raphanipes* during the drying process, resulting in the darkening of their color, referred to as browning. According to the comprehensive score of color difference in the table, it can be seen that various drying treatment methods have certain differences (*p* < 0.05). In terms of color, the *L** of VD was significantly higher than that of HD and ID, and the values of *a** and *b** were lower than those of HD and ID. It allows the sample to be in a vacuum, non-high-temperature conditions, which can reduce the possibility of enzymatic as well as non-enzymatic browning of *O. raphanipes*, and, therefore, can maximally maintain its original color. On the contrary, ID was dried at high temperature, and the samples underwent severe browning, hence the darker color of the gill. The effect of HD coloration was close to that of ID, which was affected by temperature so that it also underwent severe browning internally, which is the same as that reported by Bu et al. [39].

The microstructure of *O. raphanipes* samples after different drying treatments was observed by scanning electron microscopy. As shown in Figure 1, each sample exhibits different degrees of looseness. As can be seen at 1500 and 4000 magnification, the HD exhibits a porous, honeycomb ultrastructure with a large degree of surface crumpling and collapse. This may be related to the characteristics of hot air drying. During the heating process, the water molecules in the surface layer of *O. raphanipes* evaporate at a faster rate, while the thermal movement of water molecules in its interior is slower, so the water evaporation of *O. raphanipes* is not uniform. The organization of the HD samples corroborates with the low rehydration they exhibit. The ID samples showed a dense structure, which may help to increase the density of the ID samples; however, the collapse of the spatial structure may reduce the ability of the tissue to absorb water during rehydration. Yao et al.’s study [40] pointed out that the surface of materials obtained by infrared drying is relatively flat because infrared radiation has the same vibration frequency as the inside of the material: high heating efficiency and high heat transfer density. It can promote the rapid evaporation of water inside the object, increase the number of surface micropores, and arrange the fine cells neatly. The cellular organization of the VD samples showed a porous three-dimensional structure with only slight crumpling, large voids, loose texture, and well-maintained spatial structure. The microstructure of the material can often determine its macroscopic characteristics, so the appearance of the VD *O. raphanipes* is closer to that of the fresh product, with less shrinkage of the mushroom stem, high rehydration, and better quality.

### 3.2. Effects of Different Drying Methods on the Volatile Substances of O. raphanipes

In recent years, the HS-SPME method has been widely used for the extraction of trace amounts of volatile and semi-volatile substances. Meanwhile, gas chromatography–mass spectrometry (GC-MS) has the advantages of high sensitivity, strong qualitative ability, stability, and reliability, and is suitable for the qualitative and quantitative analysis of unknown components in multi-component mixtures. In order to evaluate the effect of different drying methods on the flavor of *O. raphanipes*, the volatile substances were determined by HS-SPME-GC-MS. There are multiple pathways for the production of volatile components: chemical or enzymatic oxidation of unsaturated fatty acids, as well as interactions with proteins, peptides, and free amino acids, degradation of long-chain compounds, and the Meladic reaction [41,42]. Total ion chromatograms of volatile compounds in *O. raphanipes* dried using different methods are shown in the Appendix A. The response intensity of volatiles in HD, ID, and VD in different drying methods is shown in Table 5. Among the three types of dried samples, 75 common volatile substances were detected, including 15 alkanes, 11 alcohols, 8 esters, 7 aldehydes, 7 ketones, 6 olefins, 6 acids, 4 benzenes, and 11 others. Alkanes are the main components of the entire volatile compound, followed by alcohols and esters.

1-octen-3-ol, known as mushroom alcohol, has a signature mushroom odor and is an aliphatic unsaturated alcohol [43]. As can be seen in Table 5, the effect of different drying techniques on the C_8_ alcohol content was significant (*p* < 0.05). The response intensity of 1-octen-3-ol was significantly different among the three samples, which is in agreement with the results reported by Luo et al. [44].

Among them, the highest response intensity of 1-octen-3-ol was found in VD, followed by ID and HD. These results suggest that C_8_ alcohols are less stable in mushrooms due to oxidization, degradation, and enzymatic reactions and are susceptible to drying temperatures and techniques, and their degradation can easily take place during heat treatments [45,46]. 1-octen-3-ol can be produced by the conversion of 13-HPOD and 10-HPOD, while 1-octen-3-ol can in turn be oxidized to 3-octanol and 3-octanone [47]. C_8_ compounds such as 1-octen-3-ol can be produced by the auto-oxidation of polyunsaturated fatty acids, enzyme-catalyzed oxidation, and cleavage of polyunsaturated fatty acids, with enzyme-catalyzed being the predominant pathway. Enzymes, being products of living cells, can interact with substrates to produce different molecular substances. Mushrooms are rich in unsaturated fatty acids (such as linoleic acid) that are oxidized by enzymes to produce eight-carbon molecules [48]. However, HD drying makes the cell integrity disrupted, which in turn affects the enzyme–substrate interaction. The response intensity of other alcohols, such as isobutanol, n-hexanol, 2,3-butanediol, 2-ethylhexanol, and nonanol, shown in Table 5, were significantly different in VD, probably due to high decarboxylase activity during ketoacid conversion [49].

Aldehydes and ketones are carbonyl compounds, of which aldehydes are a relatively abundant volatile compound in edible mushrooms, with a low odor threshold and a strong overlapping effect with other compounds. Aldehydes from C_5_ to C_9_ are usually derived from fat oxidation and degradation, and they have a fat-scented odor. From the table, it can be seen that the content of aldehydes in HD and ID was significantly higher than that in VD and concentrated in the C_5_ to C_9_ group of compounds. Among them, 3-methylbutyraldehyde and 2-methylbutyraldehyde have a cocoa flavor, and benzaldehyde has a sweet taste and a special almond flavor, which can give the product a fatty flavor. And both hexanal and benzaldehyde dimethyl acetal have a grassy odor. Ketones are used as flavor substances to provide floral and fruity aromas, with excellent and long-lasting flavor characteristics, which are mainly generated by heating through lipid oxidation [50]. The major 8c ketone in mushrooms is 3-octanone, which possesses mushroom and herbaceous flavors. As shown in Table 5, the response intensity of 3-octanone was higher after drying by hot air and infrared radiation, which may be due to the enzymatic oxidation of 3-octanol to produce 3-octanone [51]. After hot air drying and infrared radiation drying, aldehydes constitute the main aroma of *O. raphanipes* flavor characteristics, and ketones 3,5-octadien-2-one also play a certain role in contributing to providing a fruity aroma, which may be the result of heating and drying, conducive to the formation of aldehydes and ketones carbonyl compounds, whereas vacuum freeze drying cannot easily generate carbonyl flavor substances due to the low-temperature and anaerobic conditions.

Esters not only have strong volatility at room temperature and low threshold but also impart a sweet and slightly oily odor to food. As can be seen from Table 5, compared with vacuum freeze drying, the content loss of esters after hot air drying and infrared radiation drying is higher, which may be due to the reaction of ester compounds with a large amount of moisture in the drying process under heating conditions and hydrolyzed to alcohols or phenols as well as carboxyl compounds [52]. Vacuum freeze drying preserves the esters better due to the lower temperature and vacuum environment.

For hydrocarbons, more alkanes and olefins were detected in all three samples, and the response intensity of alkanes in the vacuum freeze-dried samples was higher, which may be due to the fact that with the loss of water in the samples, the volatile flavor components of olefins changed under the low-temperature and vacuum conditions, and the spatial structure changed [53], resulting in the generation of a variety of alkanes. The flavor threshold of hydrocarbons is generally high and has little effect on the overall flavor change of raw materials. However, some olefins have unique flavors, such as D-limonene, which can give a fresh orange aroma and lemon aroma to food products. In addition, longifolia alkene and Δ-juniperene can contribute herbaceous and woody aromas, respectively.

Finally, the three different drying treatments also yielded acid and benzene compounds and a range of other substances, including pyrazines, amines, furans, ethers, and naphthalenes. The response intensity of pyrazine compounds is higher after hot air drying and infrared radiation drying, which is due to the fact that infrared radiation drying makes the internal reaction of *O. raphanipes* more intense, while the high temperature of hot air drying treatment makes it undergo the Meladic reaction, which generates pyrazine compounds with a nutty flavor. In addition, 2-n-pentylfuran contributes to the flavor formation of *O. raphanipes* by providing a fruity aroma. PCA analysis was performed based on the identification results of GC-MS, and Figure 2 shows the differences between samples, with the first two principal components (PCs) accounting for 66.4% and 29.2% of the total variance, respectively, and a cumulative contribution of 95.6% for PC1 and PC2. From Figure 3, it can be further visualized that the volatile flavor substances in the *O. raphanipes* treated by the three drying methods were different from each other. The volatile compounds of Sample A were mainly concentrated in substances with higher flavor thresholds such as aldehydes and ketones, while those of Sample B were dispersed in alkanes and acids, and most of the compounds with lower flavor thresholds, such as alcohols, alkanes, and esters, were mainly concentrated in Sample C, where the differences were more significant.

### 3.3. FT-IR and UV Spectra Analysis of PS-V, PS-H and PS-I

The infrared spectral analysis of the polysaccharides obtained from the three drying treatments of *O. raphanipes* showed that there was no obvious difference between the peak positions and peak shapes of the three polysaccharides, and they had relatively similar spectral characteristics, which indicated that their structures and carbon chain skeletons were basically the same, but the peak intensities varied considerably, which indicated that the contents of their polysaccharides were different.

Polysaccharide substances usually have four typical absorption peaks. The first one is a broad absorption peak at 3416.97 cm^−1^, which is caused by the stretching vibration of O-H [54], and many hydrogen bonds can be formed between these O-H. The second is a weaker absorption peak at 2925.44 cm^−1^, which is related to the stretching vibration of the methyl or methylene groups in the polysaccharide molecule. The third is the absorption peak at 1642.56 cm^−1^, which is related to the asymmetric stretching vibration of the carbonyl group in -COOH. The fourth absorption peak is located at 1405.59 cm^−1^ and it indicates the presence of C-H variable angle vibrations in the polysaccharide. In addition, 1151.29 cm^−1^ and 1078.42 cm^−1^ are the absorption peaks of the C-O-C stretching vibration and the alcohol hydroxyl variable angle vibration, and the stronger absorption peak near 1042.70 cm^−1^ is due to the C=O vibration in the pyran ring structure [55]. Based on the fact that the polysaccharide has three absorption peaks in the interval of 1150–1010 cm^−1^, it can be hypothesized that the polysaccharide of *O. raphanipes* is not a furan-type glycocycle but a pyran-type polysaccharide, and the type of sugar is related to the number of absorption peaks in the range 596.01 cm^−1^, which is the variable angle vibrational absorption peak for β-type C-H upright bonds in sugar molecules. As shown in Figure 4B, there were weak protein absorption peaks at 280 nm for PS-V, PS-I, and PS-H (the precision of absorbance measurement is 0.01), indicating that the polysaccharides contained a small amount of proteins, which was in agreement with the results of the determination by the Caumas Brilliant Blue method.

### 3.4. Chemical Composition of PS-V, PS-H, and PS-I

From Table 6, it can be seen that compared with the other two drying methods, the highest polysaccharide yield was obtained from vacuum freeze drying *O. raphanipes* because *O. raphanipes* had a higher rehydration rate after vacuum freeze drying, and the cellular tissues of the vacuum freeze-dried samples showed a porous three-dimensional structure, which was conducive to the penetration of the extraction solvents during the extraction process. Therefore, vacuum freeze drying has a good effect in the process of extracting polysaccharides. In Table 6, the content of polysaccharides (66.77%) and glucuronic acid (20.22%) obtained after infrared radiation drying of *O. raphanipes* was higher than that obtained after vacuum freeze drying (61.50%, 16.40%) and hot air drying (60.44%, 13.20%). Infrared radiation drying disrupts the metabolic equilibrium of carbohydrates and produces macromolecules, such as polysaccharides, which increase the production of polysaccharides and glyoxalates. The ultrasonic cavitation effect and acoustic flow effect of vacuum freeze drying can cause damage to the sample, breaking down macromolecules such as polysaccharides into monosaccharides or oligosaccharides, resulting in a reduction in polysaccharide content and glyoxalate. The high temperature and long drying time of hot air drying can cause material color deterioration and nutrient degradation, and the thermal efficiency is low, so the polysaccharide content and uronic acid are relatively low. As shown in Table 6, the protein content of PS-V was 1.29%, which was slightly higher than that of PS-H and PS-I. Vacuum freeze drying has the lowest loss of protein, and the non-high-temperature vacuum environment reduces the rate of biochemical reaction in the fruiting body, which can maintain its original structure and shape and reduces the loss of nutrients. The lower protein content under hot air drying and infrared radiation drying may be due to the higher drying temperature, which promotes the decomposition reaction of proteins; in addition, the high-temperature conditions may also destroy the spatial structure of proteins and denaturation of proteins, so the degree of protein destruction is directly proportional to the drying temperature and processing time.

### 3.5. SEM Analysis of PS-V, PS-H and PS-I

Scanning electron microscopy (SEM) allows us to characterize the surface morphology of the samples and to compare the morphological differences between the samples as well as the microstructure of the polysaccharides. Figure 5 shows the morphology of polysaccharide powders extracted from different drying methods of treating *O. raphanipes* observed at 200 and 2000 electron microscopic magnifications, respectively. It can be seen that PS-H presents a loose and large lamellar structure at low magnification, and its surface can be seen to consist of a series of blocky projections at high magnification; PS-I presents a tight and smaller lamellar structure at low magnification and an irregularly arranged fragmented structure at high magnification, which may be due to the cross-linking and entanglement of polysaccharide molecules with each other. PS-V can be seen under high magnification as a reticular structure accompanied by spherical projections, indicating a large number of internal cavities, which is due to the direct sublimation of moisture from solid to gaseous states during vacuum freeze drying of the raw material without being damaged by external forces [56]. These different microscopic morphologies also indicate that the drying method affects the microstructure of polysaccharides from *O. raphanipes.*

### 3.6. Antioxidant Activities Analysis of PS-V, PS-H and PS-I

As a stabilizing free radical, DPPH is widely used to evaluate the free-radical scavenging ability of different compounds. As shown in Figure 6A, the scavenging rates of DPPH free radicals by PS-V, PS-H, and PS-I all gradually increase with their increasing mass concentrations. It can be observed that the scavenging rate of PS-I is consistently higher than the other two, ultimately reaching 70.14%, while PS-V and PS-H reach 68.78% and 63.81%, respectively. These rates are lower than the free-radical scavenging rate of the positive control, Vc, which was 99.65%. The IC50 values are 1.802, 1.306, and 0.712, respectively. The experimental results showed that the scavenging ability of DPPH radicals of IR radiation-dried polysaccharides was stronger than that of hot air drying and vacuum freeze drying, which showed good antioxidant ability.

Hydroxyl radicals, as harmful reactive oxygen species, are associated with oxidative damage to biomolecules, such as carbohydrates, proteins, lipids, and deoxyribonucleic acid in cells, leading to tissue damage and even cell death. Therefore, the removal of hydroxyl groups is very important for the conservation of organisms. As shown in Figure 6B, the scavenging rates of hydroxyl free radicals by PS-V, PS-H, and PS-I all increase with their increasing mass concentrations, and the scavenging rate of PS-I is consistently higher than the other two. When the polysaccharide mass concentration is 4.0 mg/mL, the scavenging rates under the three drying methods reach 27.61%, 31.30%, and 33.05%, respectively, which are lower than the free-radical scavenging rate of Vc, 98.71%. The IC50 values of PS-I, PS-V, and PS-H are 1.14, 0.715, and 0.268, respectively, indicating that the polysaccharide dried by infrared radiation has the strongest ability to scavenge hydroxyl free radicals. This may be due to the higher carboxyl group content of PS-I, which helps to reduce the production of hydroxyl radicals by chelating ferrous ions.

### 3.7. Inhibitory Activity Analysis of PS-V, PS-H and PS-I

As an essential enzyme in the glucose metabolism pathway in organisms, α-glucosidase is located on the surface membrane of intestinal cells, where it hydrolyzes glycosidic bonds in carbohydrates and disaccharides (e.g., sucrose, maltose) and releases glucose and other monosaccharides, which are then taken up by the intestinal epithelial cells, leading to elevated blood glucose levels [57]. Thus, the inhibition of α-glucosidase activity can effectively control postprandial blood glucose levels. As shown in Figure 7A, PS-H, PS-I, and PS-V all have a significant inhibitory effect on the activity of α-glucosidase, and this effect is dose-dependent. The inhibitory activity of PS-V is consistently higher than the other two. The IC50 values of PS-V, PS-I, and PS-H are 3.881, 2.058, and 0.973 mg·mL-1, respectively. PS-H and PS-I showed lower inhibition of α-glucosidase compared to PS-V. This may be due to the hardening of the structure due to internal moisture migration during drying, agglomeration, and possible damage to thermosensitizers and volatile components, as well as disruption of the network structure by temperature, which affects the inhibitory effect on α-glucosidase. While vacuum freeze-drying treatment inhibits the action of microorganisms and enzymes by drying at low temperatures, while maximizing the retention of the raw material’s original network structure and morphology, it has a better inhibition of α-glucosidase. Meanwhile, Nie et al. and Shen et al. [58,59] speculate that the polysaccharides have an impact on the changes in inhibition of α-glucosidase, which may be related to molecular weight distribution and uronic acid content. After degradation, polysaccharides can be absorbed by the small intestine and inhibit intestinal growth. α-glucosidase activity reduces the risk of postprandial hyperglycemia [60].

α-Amylase is, likewise, an important digestive enzyme in the breakdown of carbohydrates by the organism, and it catalyzes the cleavage of the α-D-(1–4) glycosidic bond of starch and various maltodextrins into oligosaccharides [61]. As shown in Figure 7B, PS-H, PS-I, and PS-V also exhibit dose-dependent inhibitory activity on α-amylase. The inhibitory rate of PS-V at all concentrations is higher than the other two. The IC50 values of PS-V, PS-I, and PS-H are 0.938, 0.685, and 0.503 mg·mL^−1^, respectively. In general, polysaccharides may inhibit starch decomposition after entering the body in two ways: First, the polysaccharides may adsorb onto the starch, thereby hindering the decomposition of the starch. Second, some groups in polysaccharides, such as carboxyl groups, may interact with amino acid residues in α-amylase through hydrogen bonding, thus forming a complex between polysaccharides and α-amylase, which in turn affects the spatial conformation of α-amylase [62]. From the figure, it can be seen that PS-V has higher α-amylase inhibition, which may be due to the fact that the vacuum freeze-drying treatment better maintains the original structure of the polysaccharide to maximize the encapsulation of the enzyme or starch, whereas the structure of the polysaccharide treated by hot air drying and radiation drying degraded during the drying process, which affects the inhibition of α-amylase activity.

## 4. Conclusions

This study showed that the sensory and nutritional differences between *O. raphanipes* obtained using three different drying methods were more significant. Dried products of *O. raphanipes* treated by vacuum freeze drying had the highest rehydration rate and were closer in appearance quality to the fresh samples, but they were lower than hot air drying in terms of flavor scores, which was consistent with the results of the subsequent volatiles assay. The microstructure of *O. raphanipes* dried by hot air drying and infrared radiation was obviously changed, while vacuum freeze drying could maintain the porous spatial three-dimensional structure of the mushroom. Alkanes are the main components of the overall volatile compounds, followed by alcohols and esters. The relative content of vacuum freeze-dried volatile substances is higher because some volatile substances are easily degraded or decomposed during heating and infrared radiation treatment, and vacuum freeze drying can reduce the occurrence of this reaction. PS-V had the highest yield among the polysaccharides mentioned in the three dried products, and the structures of the three were basically the same, but there were some differences in the chemical compositions: PS-V could be seen as a reticulated structure accompanied by globular protrusions under high magnification, suggesting that there was a large number of internal cavities, whereas PS-H and PS-I showed a lamellar structure in general. The antioxidant activities of the three polysaccharides against DPPH and ·OH as well as the inhibitory activities against α-glucosidase and α-amylase, although all of them were reduced to varying degrees, PS-V exhibited stronger biological activities compared to PS-H and PS-I. In conclusion, if the main consideration is the appearance quality and nutrient retention of *O. raphanipes* after drying, vacuum freeze drying is the best; however, if considering the cost and energy consumption of the product drying process, hot air drying is recommended because it is simple to operate, short in time, low in energy consumption, and also has a high flavor score. Infrared radiation drying is in between the two and can be chosen differently according to needs. This provides a theoretical basis for the selection of the drying method of *O. raphanipes* and could be used to guide the practical processing of *O. raphanipes.*

## Figures and Tables

**Figure 1 foods-13-01087-f001:**
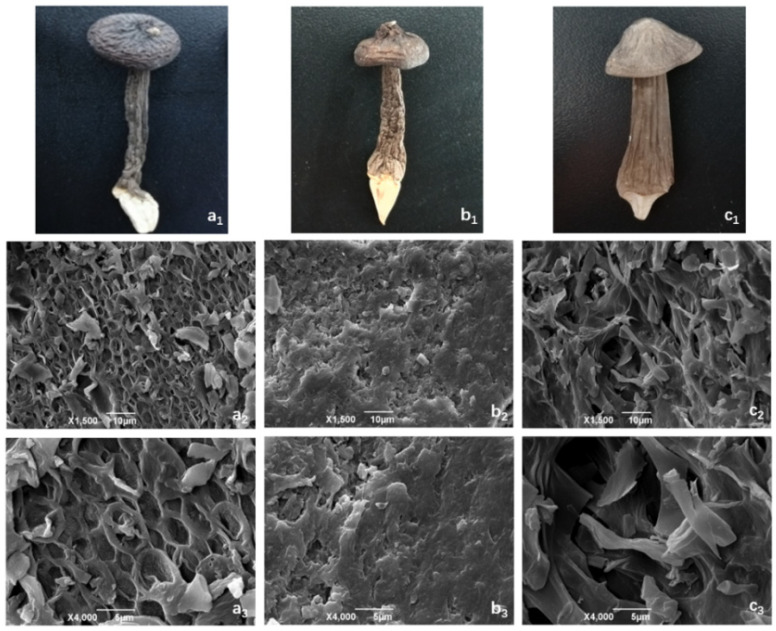
Effect of different drying methods on the microstructure of *O. raphanipes*: (**a**) hot air drying, (**b**) infrared radiation drying, (**c**) vacuum freeze drying. (**1**) Appearance, (**2**) the magnification was set as ×1500, (**3**) the magnification was set as ×4000. (**a_1_**,**b_1_**,**c_1_**) are the appearance of samples treated with hot air drying, infrared radiation drying and vacuum freeze drying, respectively; (**a_2_**,**b_2_**,**c_2_**) are the magnification was set as ×1500 of samples treated with hot air drying, infrared radiation drying and vacuum freeze drying, respectively; (**a_3_**,**b_3_**,**c_3_**) are the magnification was set as ×4000 of samples treated with hot air drying, infrared radiation drying and vacuum freeze drying, respectively.

**Figure 2 foods-13-01087-f002:**
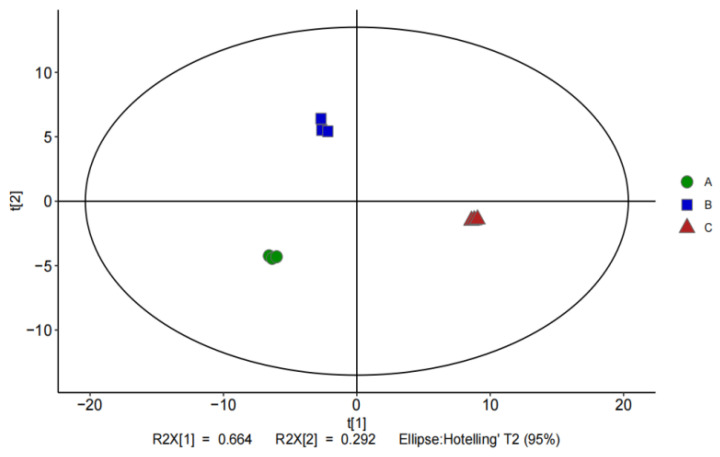
PCA analysis of volatile compounds in *O. raphanipes* dried using different methods, (A) HD-treated samples, (B) ID-treated samples, (C) VD-treated samples.

**Figure 3 foods-13-01087-f003:**
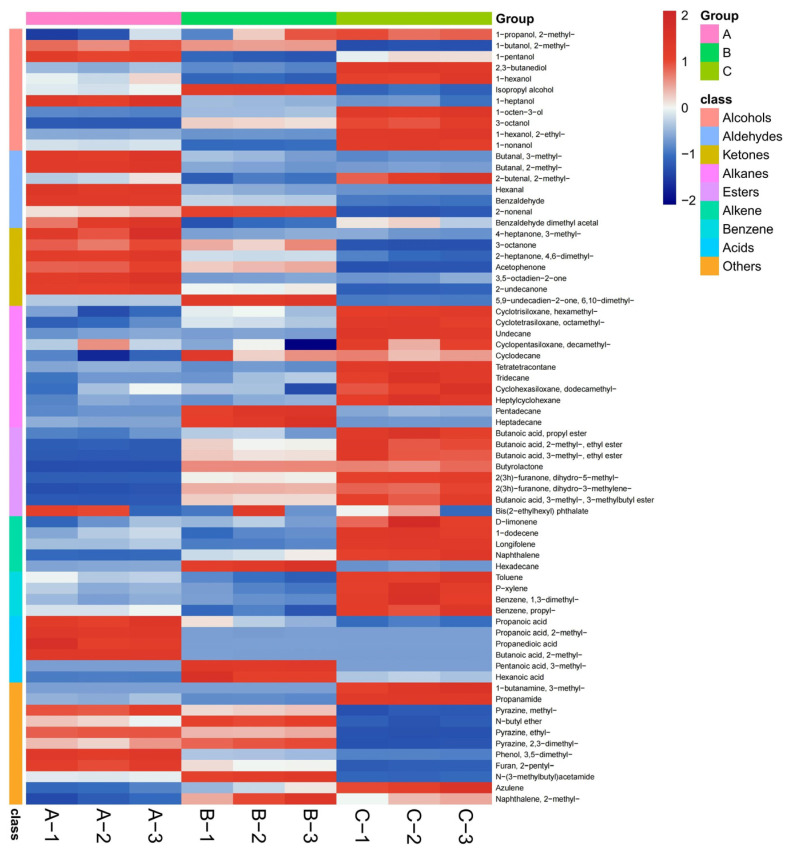
Heatmap analysis of volatile compounds in *O. raphanipes* dried using different methods, (A) HD-treated samples, (B) ID-treated samples, (C) VD-treated samples.

**Figure 4 foods-13-01087-f004:**
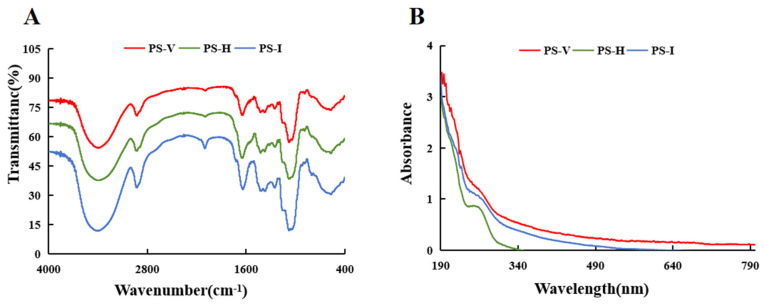
(**A**) Infrared spectrum of PS-V, PS-H and PS-I; (**B**) ultraviolet spectrum of PS-V, PS-H, and PS-I.

**Figure 5 foods-13-01087-f005:**
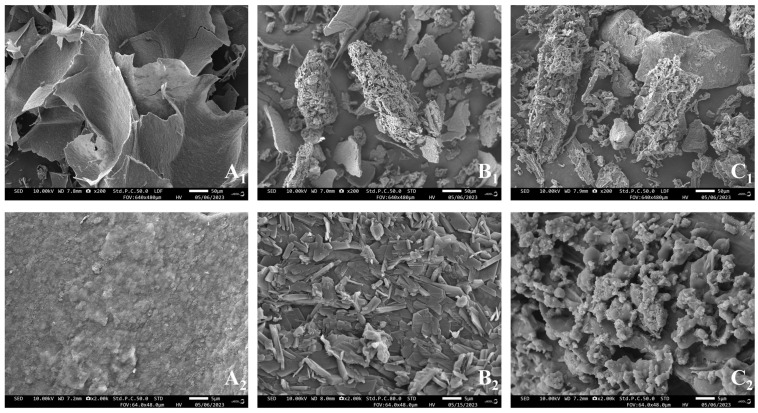
Effect of different drying methods on the microstructure of *O. raphanipes* polysacchaaride, (**A**) PS-H, (**B**) PS-I, (**C**) PS-V. (**1**) The magnification was set as ×200; (**2**) the magnification was set as ×2000.

**Figure 6 foods-13-01087-f006:**
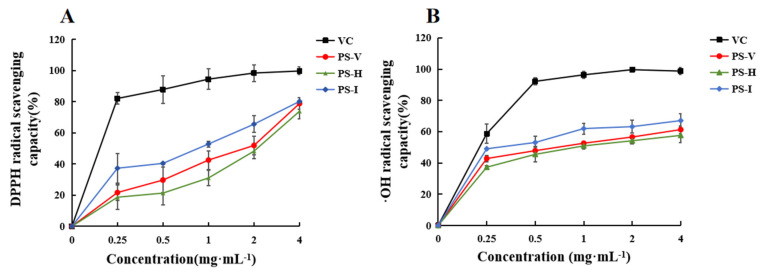
Effect of different drying methods on vitro antioxidant activity of polysaccharides, (**A**) DPPH radical scavenging capacity and (**B**) ·OH radical scavenging capacity. Data are expressed as the mean ± standard deviation (*n* = 3).

**Figure 7 foods-13-01087-f007:**
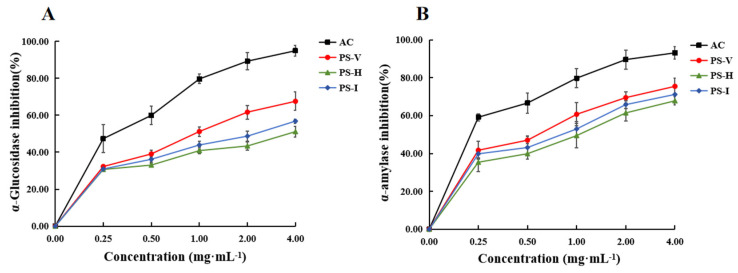
Effect of different drying methods on enzyme inhibitory activity of polysaccharides, (**A**) α-glucosidase and (**B**) α-amylase. Data are expressed as the mean ± standard deviation (*n* = 3).

**Table 1 foods-13-01087-t001:** Different drying conditions.

Method	Sample Weight	Temperature	Parameter Settings
HD	50 g	50 °C	Convection rate: 0.3 m^3^/s
ID	50 g	50 °C	Infrared intensity: 2 W/cm^2^
			Infrared distance: 10 cm
			Wind speed: 1.5 m/s
VD	50 g	−20 °C and −80 °C	Cold trap temperature: −50 °C
			Vacuum degree: 1.5 hPa

HD: hot air drying; ID: infrared radiation drying; VD: vacuum freeze drying.

**Table 2 foods-13-01087-t002:** Appearance quality evaluation standard.

Evaluation Indicators	Rating Scale					Weighted Value			
Gill color	white	4	Light yellow	3	yellow	2	Deep yellow	1	3.5
Flavor	Strong	4	normal	3	less	2	none	1	3.5
Degree of shrinkage	No shrinkage	4	Thicker shank	3	thick-shank	2	thin-shank	1	3

**Table 3 foods-13-01087-t003:** Effect of different drying methods on appearance quality evaluation.

Drying Method	Scoring Indicators			Totals
	Gill Color	Flavor	Degree of Shrinkage	
HD	8.4	11.9	4.8	25.1
ID	10.2	9.8	5.7	25.7
VD	12.9	8.8	10.5	32.2

HD: hot air drying; ID: infrared radiation drying; VD: vacuum freeze drying.

**Table 4 foods-13-01087-t004:** Effect of different drying methods on moisture content, rehydration ratio, and surface color.

Drying Method	Drying Time (h)	Moisture Content (%)	Rehydration Rate (%)	Color			
				*L**	*a**	*b**	ΔE
Fresh	-	-	-	93.11 ± 0.08 ^a^	0.65 ± 0.03 ^d^	6.46 ± 0.68 ^c^	-
HD	10	6.5 ± 0.1	3.96 ± 0.15 ^b^	66.68 ± 0.20 ^c^	2.45 ± 0.05 ^b^	11.69 ± 0.08 ^a^	27.13 ± 0.04 ^b^
ID	10	6.5 ± 0.1	4.11 ± 0.21 ^b^	65.17 ± 0.08 ^d^	2.66 ± 0.02 ^a^	12.30 ± 0.13 ^a^	28.74 ± 0.20 ^a^
VD	24	6.5 ± 0.1	6.70 ± 0.17 ^a^	79.26 ± 0.04 ^b^	0.60 ± 0.04 ^c^	7.96 ± 0.12 ^b^	14.00 ± 0.05 ^c^

HD: hot air drying; ID: infrared radiation drying; VD: vacuum freeze drying. Here, *L** represents brightness, while *a** and *b** represent chroma, corresponding to the opposing colors of red/green and yellow/blue, respectively. The formula for calculating color difference is: ΔE = $\sqrt{{\left(L_{2}-L_{1}\right)}^{2}+{\left(a_{2}-a_{1}\right)}^{2}+{\left(b_{2}-b_{1}\right)}^{2}}$, where (*L*_1_, *a*_1_, *b*_1_) and (*L*_2_, *a*_2_, *b*_2_) are the lab coordinates of the two colors. Data are expressed as the mean ± standard deviation (*n* = 3), and different letters in the same column indicate significant differences (*p* < 0.05).

**Table 5 foods-13-01087-t005:** The content of volatile compounds in *Oudemansiella raphanipes* dried using different methods.

Volatile Compounds	Rt (min)	Formula	CAS	Odorant Description ^a^	Response Intensity		
					HD	ID	VD
Alcohols							
1-propanol, 2-methyl-	2.06	C_4_H_10_O	78-83-1	ethereal	1.10 ± 0.26 ^b^	1.50 ± 0.33 ^ab^	1.78 ± 0.04 ^a^
1-butanol, 2-methyl-	3.206	C_5_H_12_O	137-32-6	ethereal	24.04 ± 1.56 ^a^	21.60 ± 0.28 ^b^	4.30 ± 0.27 ^c^
1-pentanol	3.67	C_5_H_12_O	71-41-0	Fermented, fusel	26.67 ± 0.60 ^a^	15.08 ± 0.44 ^c^	21.51 ± 0.62 ^b^
2,3-butanediol	4.148	C_4_H_10_O_2_	513-85-9	creamy	4.67 ± 0.23 ^b^	3.73 ± 0.12 ^c^	9.55 ± 0.15 ^a^
1-hexanol	5.818	C_6_H_14_O	111-27-3	pungent, ethereal	10.72 ± 0.58 ^b^	7.75 ± 0.16 ^c^	13.95 ± 0.50 ^a^
Isopropyl alcohol	6.247	C_3_H_8_O	67-63-0	alcoholic	29.21 ± 1.35 ^b^	52.61 ± 1.79 ^a^	11.19 ± 1.57 ^c^
1-heptanol	8.551	C_7_H_16_O	111-70-6	fermented	4.25 ± 0.23 ^a^	2.37 ± 0.05 ^b^	2.01 ± 0.17 ^c^
1-octen-3-ol	8.793	C_8_H_16_O	3391-86-4	mushroom	83.52 ± 1.38 ^c^	103.39 ± 1.56 ^b^	185.08 ± 5.51 ^a^
3-octanol	9.257	C_8_H_18_O	589-98-0	mushroom, nutty,	0.93 ± 0.04 ^c^	7.99 ± 0.23 ^b^	12.30 ± 0.71 ^a^
1-hexanol, 2-ethyl-	10.236	C_8_H_18_O	104-76-7	citrus	11.37 ± 0.47 ^b^	6.96 ± 0.29 ^c^	61.86 ± 0.76 ^a^
1-nonanol	14.378	C_9_H_20_O	143-08-8	fresh, clean, fatty	8.87 ± 0.13 ^b^	4.69 ± 0.09 ^c^	15.62 ± 0.33 ^a^
Aldehydes							
Butanal, 3-methyl-	2.269	C_5_H_10_O	590-86-3	fruity, cocoa	24.37 ± 0.84 ^a^	12.95 ± 0.86 ^b^	11.24 ± 0.28 ^c^
Butanal, 2-methyl-	2.349	C_5_H_10_O	96-17-3	cocoa	36.03 ± 0.78 ^a^	14.83 ± 0.74 ^b^	14.71 ± 0.24 ^b^
2-butenal, 2-methyl-	3.288	C_5_H_8_O	1115-11-3	green, fruity	7.14 ± 0.33 ^b^	5.91 ± 0.19 ^c^	9.01 ± 0.46 ^a^
Hexanal	4.236	C_6_H_12_O	66-25-1	green	43.13 ± 1.57 ^a^	4.15 ± 1.69 ^b^	0.13 ± 0.12 ^c^
Benzaldehyde	8.22	C_7_H_6_O	100-52-7	sweet, almond	106.52 ± 1.57 ^a^	39.45 ± 2.05 ^b^	10.62 ± 1.29 ^c^
2-nonenal	11.504	C_9_H_16_O	2463-53-8	fatty	7.50 ± 0.32 ^b^	9.62 ± 0.06 ^a^	3.44 ± 0.09 ^c^
Benzaldehyde dimethyl acetal	12.613	C_9_H_12_O_2_	1125-88-8	green	233.26 ± 22.22 ^a^	90.23 ± 9.54 ^c^	161.52 ± 17.78 ^b^
Ketones							
4-heptanone, 3-methyl-	8.521	C_8_H_16_O	15726-15-5	/	2.81 ± 0.21 ^a^	1.75 ± 0.03 ^b^	1.71 ± 0.06 ^b^
3-octanone	9.007	C_8_H_16_O	106-68-3	mushroom, herbal	71.54 ± 2.53 ^a^	63.64 ± 3.79 ^b^	33.04 ± 0.48 ^c^
2-heptanone, 4,6-dimethyl-	10.766	C_9_H_18_O	19549-80-5	/	5.72 ± 0.20 ^a^	3.43 ± 0.02 ^b^	2.12 ± 0.20 ^c^
Acetophenone	11.313	C_8_H_8_O	98-86-2	floral	17.32 ± 0.60 ^a^	14.32 ± 0.49 ^b^	5.86 ± 0.08 ^c^
3,5-octadien-2-one	11.452	C_8_H_12_O	30086-02-3	fruity	91.73 ± 3.11 ^a^	53.39 ± 0.93 ^b^	52.99 ± 1.92 ^b^
2-undecanone	17.788	C_11_H_22_O	112-12-9	fruity	120.11 ± 2.94 ^a^	73.65 ± 2.47 ^b^	26.70 ± 0.99 ^c^
5,9-undecadien-2-one, 6,10-dimethyl-	21.909	C_13_H_22_O	3796-70-1	floral	16.12 ± 0.27 ^b^	49.35 ± 1.65 ^a^	4.14 ± 0.17 ^c^
Alkanes							
Cyclotrisiloxane, hexamethyl-	4.722	C_6_H_18_O_3_Si_3_	541-05-9	/	316.79 ± 20.64 ^c^	366.56 ± 13.93 ^b^	446.81 ± 4.24 ^a^
Cyclotetrasiloxane, octamethyl-	9.492	C_8_H_24_O_4_Si_4_	556-67-2	/	16.65 ± 1.22 ^c^	21.60 ± 0.76 ^b^	30.90 ± 0.78 ^a^
Undecane	12.298	C_11_H_24_	1120-21-4	/	56.56 ± 3.29 ^b^	56.52 ± 1.09 ^b^	131.53 ± 2.58 ^a^
Cyclopentasiloxane, decamethyl-	14.006	C_10_H_30_O_5_Si_5_	541-02-6	/	269.57 ± 12.65 ^ab^	247.61 ± 26.99 ^b^	291.65 ± 10.13 ^a^
Cyclodecane	14.369	C_10_H_20_	293-96-9	/	0.79 ± 0.22 ^b^	1.80 ± 0.30 ^a^	1.69 ± 0.09 ^a^
Tetratetracontane	17.411	C_44_H_90_	7098-22-8	/	10.48 ± 0.26 ^b^	9.39 ± 0.35 ^c^	20.53 ± 0.34 ^a^
Tridecane	17.951	C_13_H_28_	629-50-5	/	155.38 ± 12.39 ^b^	178.86 ± 16.75 ^b^	317.16 ± 15.55 ^a^
Cyclohexasiloxane, dodecamethyl-	18.786	C_12_H_36_O_6_Si_6_	540-97-6	/	562.40 ± 35.31 ^b^	546.07 ± 39.05 ^b^	680.09 ± 23.75 ^a^
Heptylcyclohexane	19.06	C_13_H_26_	5617-41-4	/	9.10 ± 0.64 ^b^	9.45 ± 0.56 ^b^	17.58 ± 0.81 ^a^
Pentadecane	23.048	C_15_H_32_	629-62-9	waxy	7.66 ± 0.35 ^b^	20.59 ± 1.42 ^a^	9.08 ± 0.38 ^b^
Heptadecane	27.629	C_17_H_36_	629-78-7	/	2.87 ± 0.18 ^b^	10.75 ± 0.99 ^a^	2.39 ± 0.06 ^b^
Esters							
Butanoic acid, propyl ester	4.63	C_7_H_14_O_2_	105-66-8	fruity	0.59 ± 0.08 ^c^	0.90 ± 0.17 ^b^	2.19 ± 0.15 ^a^
Butanoic acid, 2-methyl-, ethyl ester	5.348	C_7_H_14_O_2_	7452-79-1	fruity	0.29 ± 0.02 ^c^	1.21 ± 0.08 ^b^	1.92 ± 0.18 ^a^
Butanoic acid, 3-methyl-, ethyl ester	5.424	C_7_H_14_O_2_	108-64-5	fruity	0.75 ± 0.07 ^c^	8.09 ± 0.42 ^b^	12.63 ± 1.25 ^a^
Butyrolactone	6.985	C_4_H_6_O_2_	96-48-0	creamy, milky	1.68 ± 0.03 ^b^	19.15 ± 0.09 ^a^	19.73 ± 0.91 ^a^
2(3h)-furanone, dihydro-5-methyl-	8.035	C_5_H_8_O_2_	108-29-2	herbal	3.56 ± 0.14 ^c^	29.01 ± 0.47 ^b^	50.99 ± 1.46 ^a^
2(3h)-furanone, dihydro-3-methylene-	10.388	C_5_H_6_O_2_	547-65-9	/	78.74 ± 2.16 ^c^	204.18 ± 1.29 ^b^	240.12 ± 8.42 ^a^
Butanoic acid, 3-methyl-, 3-methylbutyl ester	12.475	C_10_H_20_O_2_	659-70-1	fruity	0.77 ± 0.08 ^c^	12.04 ± 0.35 ^b^	18.82 ± 1.10 ^a^
Bis(2-ethylhexyl) phthalate	34.429	C_24_H_38_O_4_	117-81-7	/	0.06 ± 0.02 ^a^	0.07 ± 0.02 ^a^	0.06 ± 0.02 ^a^
Alkene							
D-limonene	10.182	C_10_H_16_	5989-27-5	citrus	13.73 ± 1.01 ^b^	14.51 ± 0.55 ^b^	19.59 ± 1.52 ^a^
1-dodecene	14.947	C_12_H_24_	112-41-4	/	3.52 ± 0.19 ^b^	3.02 ± 0.15 ^a^	5.18 ± 0.11 ^a^
Longifolene	20.774	C_15_H_24_	475-20-7	woody	1.73 ± 0.04 ^b^	1.43 ± 0.03 ^c^	2.78 ± 0.03 ^a^
Naphthalene	23.676	C_15_H_24_	483-76-1	herbal	4.50 ± 0.03 ^c^	6.93 ± 0.38 ^b^	10.18 ± 0.42 ^a^
Hexadecane	25.397	C_16_H_32_	629-73-2	/	9.88 ± 0.22 ^b^	29.17 ± 1.78 ^a^	8.79 ± 0.47 ^b^
Benzene							
Toluene	3.657	C_7_H_8_	108-88-3	sweet	16.26 ± 0.72 ^b^	12.78 ± 0.78 ^c^	22.63 ± 0.86 ^a^
P-xylene	5.786	C_8_H_10_	106-42-3	/	38.97 ± 2.07 ^b^	33.17 ± 2.30 ^c^	65.93 ± 3.90 ^a^
Benzene, 1,3-dimethyl-	5.591	C_8_H_10_	108-38-3	/	11.50 ± 0.76 ^b^	9.96 ± 0.65 ^b^	26.63 ± 2.11 ^a^
Benzene, propyl-	8.021	C_9_H_12_	103-65-1	/	1.18 ± 0.02 ^b^	0.96 ± 0.04 ^c^	1.47 ± 0.05 ^a^
**Acids**							
Propanoic acid	2.876	C_3_H_6_O_2_	79-09-04	dairy, fruity	11.52 ± 0.82 ^a^	4.27 ± 1.63 ^b^	0.48 ± 0.38 ^c^
Propanoic acid, 2-methyl-	3.997	C_4_H_8_O_2_	79-31-2	acidic	61.07 ± 0.75 ^a^	3.57 ± 0.41 ^b^	3.82 ± 0.02 ^b^
Propanedioic acid	6.352	C_3_H_4_O_4_	141-82-2	/	184.53 ± 29.51 ^a^	0.06 ± 0.02 ^b^	0.03 ± 0.01 ^b^
Butanoic acid, 2-methyl-	6.523	C_5_H_10_O_2_	116-53-0	fruity	132.86 ± 0.78 ^a^	0.37 ± 0.42 ^b^	0.10 ± 0.04 ^b^
Pentanoic acid, 3-methyl-	6.625	C_6_H_12_O_2_	105-43-1	sour, fresh	0.05 ± 0.03 ^b^	265.47 ± 2.16 ^a^	0.18 ± 0.07 ^b^
Hexanoic acid	7.178	C_6_H_12_O_2_	142-62-1	cheesy,	0.02 ± 0.01 ^c^	2.91 ± 0.37 ^a^	0.78 ± 0.08 ^b^
Others							
1-butanamine, 3-methyl-	2.781	C_5_H_13_N	107-85-7	/	0.16 ± 0.12 ^b^	0.17 ± 0.13 ^b^	226.31 ± 27.54 ^a^
Propanamide	3.954	C_3_H_7_NO	79-05-0	/	20.97 ± 1.71 ^b^	14.87 ± 0.43 ^c^	56.50 ± 0.65 ^a^
Pyrazine, methyl-	4.723	C_5_H_6_N_2_	109-08-0	nutty	116.40 ± 5.67 ^a^	86.89 ± 2.35 ^b^	31.05 ± 2.04 ^c^
N-butyl ether	6.137	C_8_H_18_O	142-96-1	ethereal	7.29 ± 0.69 ^b^	11.01 ± 0.25 ^a^	1.68 ± 0.24 ^c^
Pyrazine, ethyl-	6.952	C_6_H_8_N_2_	13925-00-3	nutty	45.40 ± 0.44 ^a^	36.69 ± 0.70 ^b^	8.54 ± 0.11 ^c^
Pyrazine, 2,3-dimethyl-	7.052	C_6_H_8_N_2_	5910-89-4	nutty	48.62 ± 4.31 ^b^	62.15 ± 1.93 ^a^	8.17 ± 0.48 ^c^
Phenol, 3,5-dimethyl-	7.517	C_8_H_10_O	108-68-9	balsamic	8.24 ± 0.29 ^a^	2.59 ± 0.15 ^b^	1.00 ± 0.03 ^c^
Furan, 2-pentyl-	9.122	C_9_H_14_O	3777-69-3	fruity	122.00 ± 4.71 ^a^	79.73 ± 3.22 ^b^	29.80 ± 0.70 ^c^
N-(3-methylbutyl)acetamide	13.482	C_7_H_15_NO	13434-12-3	/	123.40 ± 2.52 ^b^	237.75 ± 10.85 ^a^	33.20 ± 1.57 ^c^
Azulene	14.681	C_10_H_8_	275-51-4	/	22.15 ± 0.52 ^c^	25.83 ± 1.34 ^b^	32.38 ± 1.29 ^a^
Naphthalene, 2-methyl-	18.214	C_11_H_10_	91-57-6	floral	2.15 ± 0.07 ^c^	3.15 ± 0.22 ^a^	2.83 ± 0.11 ^b^

“/”, not found. HD: hot air drying; ID: infrared radiation drying; VD: vacuum freeze drying. Data are expressed as the mean ± standard deviation (*n* = 3), and different letters within a row are significantly different (*p* < 0.05). ^a^ Odorant descriptions were obtained from the Good Scents Company Information System.

**Table 6 foods-13-01087-t006:** Effect of different drying methods on general physicochemical properties.

Samples	PS-V	PS-I	PS-H
Yield (%)	1.92	1.74	1.67
polysaccharide content (wt%)	61.50 ± 1.45 ^b^	66.77 ± 1.77 ^a^	60.44 ± 1.79 ^b^
protein (wt%)	1.29 ± 0.05 ^a^	1.09 ± 0.08 ^b^	1.06 ± 0.13 ^b^
Uronic acid (wt%)	16.04 ± 0.46 ^b^	20.22 ± 0.99 ^a^	13.20 ± 0.53 ^c^
Polyphenols (wt%)	5.30 ± 0.34 ^a^	4.56 ± 0.24 ^b^	3.29 ± 0.19 ^c^

Data are expressed as the mean ± standard deviation (*n* = 3), and different letters within a row are significantly different (*p* < 0.05).

## Data Availability

The original contributions presented in the study are included in the article/Appendix A, further inquiries can be directed to the corresponding author.

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
