# Peer review of "Effect of Different Drying Methods on the Quality of Oudemansiella raphanipes"

_foods, 2024, doi:10.3390/foods13071087_

Round 1
Reviewer 1 Report
Comments and Suggestions for Authors
The manuscript titled “Effect of different drying methods on the quality of Oudemansiella raphanipes” presents the influence of the drying method on selected quality parameters of Oudemansiella raphanipes. Modern methods were used to present changes occurring during drying. Drying is one of the methods of food preservation and for this purpose this process is carried out not for the sake of drying alone. There are no storage tests aimed at drying. Moreover, even though the work is valuable in terms of analyzes performed, it was not done carefully.
The introduction lacks a reference to literature data being the subject of study.
The research material was very sparsely described. Material preparation was not described.
Research methods are insufficient, described for example, the lack of devices used with the manufacturer, in the country.
Freeze-drying – what happened to the dried material during the two-hour interval. 13% moisture in the freeze-dried product? Very high
Line 159 „10” in words
Lines 167-170 – describe the method in detail, equipment, number of repetitions, color parameters, are written in italics
Line 186 – powder??? As received
Line 257 concentration not recorded correctly, no explanation EP
Chapter 2.5 no description of the modification or method
Line 282 Lack of relevant information to enable the analysis to be reproduced
Table 3 why was there such a different moisture in the methodology. There is no description of color parameters ΔE in the methodology or in the table
„n” „P” “L*” “a*” “b*” are written in italics
Figure 2 she did not understand at all what volatile compounds it concerned
Figure 3 illegible
Figure 6 i 7 not described accurately
Author Response
Dear Reviewer and Editor:
We greatly appreciate your time and patience in conducting a professional review of our paper. We are grateful for your valuable suggestions, which have played a significant role in improving our paper. We have thoroughly studied each comment and made corresponding modifications to the paper. Below are our responses to each comment.
Comments1: The introduction lacks a reference to literature data being the subject of study.
Response1: Thank you for pointing out the deficiencies. Relevant literature data related to the research topic has been inserted into the introduction and highlighted."
Comments2: The research material was very sparsely described. Material preparation was not described.
Response2: Thank you for your valuable suggestions. Descriptions of the research materials have been added to the introduction and highlighted. Descriptions of material preparation have been added to Section 2.1 and highlighted
Comments3: Research methods are insufficient, described for example, the lack of devices used with the manufacturer, in the country.
Response3: Thank you for your suggestions. The equipment involved in the experiment, along with their manufacturers and countries, have been added to Section 2.1 and highlighted.
Comments4: Freeze-drying – what happened to the dried material during the two-hour interval. 13% moisture in the freeze-dried product? Very high
Response4: Thank you for your question. During the freeze-drying process, the moisture content of the raw material will continuously decrease. We measure the moisture content every two hours until it drops below 13%, at which point we stop the freeze-drying. The 13% moisture content here is determined based on the raw material being edible fungi. In natural or artificial conditions, the process of promoting the evaporation of moisture in fresh edible fungi to reduce the moisture content to below 13% is referred to as the drying of edible fungi. Most microorganisms require a water activity above 0.6 to grow. There is a direct relationship between the water content of food and its water activity. When the water content of food is reduced to about 13%, its water activity is usually below 0.6, effectively preventing the growth of microorganisms. At the same time, appropriate drying can maintain the nutritional components and taste of the food. However, excessive drying may lead to a decline in the taste of the food and the loss of nutritional components; if the water content is too high, it may lead to food spoilage. A water content of 13% can usually achieve the effect of maintaining food quality and extending shelf life.
Comments5: Line 159 „10” in words
Response5: Thank you for pointing out the error. The correction has been made in Section 2.3.1 of the document, and the changes have been highlighted.
Comments6: Lines 167-170 – describe the method in detail, equipment, number of repetitions, color parameters, are written in italics
Response6: Thank you for pointing out the shortcomings. Detailed descriptions have been added to the color difference determination in section 2.3.1, and the color parameters have been changed to italic writing. The revised content is highlighted.
Comments7: Line 186 – powder??? As received
Response7: Thank you for your question. The dried samples need to be observed with a scanning electron microscope (SEM), and the preprocessing requires the samples to be ground into powder. This can significantly increase their surface area, allowing more areas to be scanned by the electron beam. Also, during SEM observation, a thin layer of metal film is usually coated to prevent charge accumulation. Grinding the samples into powder can make the coating more uniform, thereby obtaining better image quality. Finally, for the dried O. raphanipes, as a solid material, its internal structure may differ from the surface structure. Grinding the sample into powder can reveal its internal structure, providing more comprehensive information.
Comments8: Line 257 concentration not recorded correctly, no explanation EP
Response8: Thank you for your question. The concentration in section 2.8.1 has been modified and highlighted. Here, 'EP' refers to Eppendorf tubes, which are typically made of polypropylene and are commonly used laboratory equipment. Here, a 4ml solution needs to be prepared, and to avoid omission, a 10ml EP tube was used for the experiment. I apologize for our oversight in writing. The full name has been added and highlighted.
Comments9: Chapter 2.5 no description of the modification or method
Response9: Thank you for pointing out the deficiencies. The specific method for polysaccharide extraction has been added to section 2.5 and highlighted."
Comments10: Line 282 Lack of relevant information to enable the analysis to be reproduced
Response10:We are very sorry, perhaps due to the version of the paper, we were unable to find the error you pointed out on line 282. We value your valuable suggestions. If you could specify in which section it is, we will seriously make modifications. Thank you very much for your patience and understanding.
Comments11: Table 3 why was there such a different moisture in the methodology. There is no description of color parameters ΔE in the methodology or in the table
Response11: Thank you for pointing out the issue. The experiment requires that the final moisture content of the dried samples must be below 13% to achieve the optimal drying effect for edible fungi. Through experiments, it was found that the lowest moisture content that far-infrared drying can achieve is around 6%. To reduce errors in subsequent experiments, the final moisture content of all three dried samples was controlled at this level. Descriptions of color and ΔE have been added to Table 3."
Comments12: “n” “P” “L*” “a*” “b*” are written in italics
Response12: Thank you for pointing out the mistakes. The "L*", "a*", "b*" in Table 3 and "n", "P" in the paper have been italicized and highlighted.
Comments13: Figure 2 she did not understand at all what volatile compounds it concerned
Response13: Thank you for pointing out the issue. The volatile compounds mentioned here refer to the substances involved in Table 4. A, B, and C represent three types of dried samples. We performed PCA (Principal Component Analysis) on them in order to more clearly distinguish the differences among the volatile compounds of the three dried samples.
Comments14: Figure 3 illegible
Response14: Thank you for pointing out the shortcomings. In Figure 3, the horizontal axis represents the sample names, and the vertical axis represents the different volatile substances (filtered according to VIP>1, P<0.05). The color changes from blue to red indicate that the response intensity of volatile substances increases, that is, the redder it is, the higher the response intensity of the volatile substances.The Figure 3 has been replaced with a high-resolution image.
Comments15: Figure 6 i 7 not described accurately
Response15: Thank you for pointing out the deficiencies. We have made supplements and modifications to the descriptions of Figure 6 and Figure 7 in Sections 3.6 and 3.7.
Thanks for your professional suggestions.We tried our best to improve the manuscript and made some changes to the manuscript. Hope that thecorrection will meet with approval. Once again thank you very much for your comments and suggestions.
With kind regards,
Yours sincerely,
Ms. Hou

Reviewer 2 Report
Comments and Suggestions for Authors
Hou et al., have studied the effect of three drying techniques on the physicochemical characteristics of dried Oudemansiella raphanipes and their extracts. Their results showed vacuum drying method yields the best qualitative and structural features in the dried products, and also the best biochemical markers in the extracts. They have conducted a detailed study, but some revisions are needed.
1. More details about the hot drying method, such as the convection rate is needed.
2. Representative HS-SPME-GC-MS spectrograms of the 3 drying methods should be shown either in the manuscript or in supplementary information. The data reduction process has to be clarified. It is not clear to me what the relative content % in Table 4 mean as some of the values are higher than 100%.
3. What is the effect of using 230C for extraction in the HS-SPME-GC-MS procedure? I think that will affect the compositions severely. Use of such high T for extraction has to be justified.
4. Any comment on why toxic chemicals such as aromatic benzene/toluene/xylene content is high in the dried mushrooms? Why do vacuum dried mushrooms leach out more aromatic hydrocarbons than the other two?
5. As per Table 2, VD samples are lighter in color than the HD and ID samples. Solutions of polysaccharides extracted from these samples seem to show the opposite trend – PS-VD has higher absorbance throughout the UV-vis range indicating the solutions with PS-VD are more colored. Can the authors comment on this?
6. Absorbance values of >2 are error prone as only <1% of the light is transmitted. Authors should report the precision of their absorbance measurements.
7. Authors should comment on the choice of drying temperature of 60C. I assume there is an interaction between temperature, time, and drying method.
Comments on the Quality of English Language
There are many compound sentences splitting which would improve the readability of the manuscript. In some places incorrect punctuation is used.
Author Response
Dear Reviewer and Editor:
We greatly appreciate the time you've taken to review our paper, titled "Effect of different drying methods on the quality of Oudemansiella raphanipes". We are grateful for the beneficial suggestions from the reviewers, which have played a crucial role in improving our paper. We have carefully examined each comment and made corresponding modifications to the paper. Below are our point-by-point responses to the reviewers' comments.
Comments1: More details about the hot drying method, such as the convection rate is needed.
Response1: Thank you for pointing out the issue. We have added the convection rate (0.3m³/s) of the hot drying method in Section 2.2, and highlighted it.
Comments2: Representative HS-SPME-GC-MS spectrograms of the 3 drying methods should be shown either in the manuscript or in supplementary information. The data reduction process has to be clarified. It is not clear to me what the relative content % in Table 4 mean as some of the values are higher than 100%.
Response2:Thank you for your guidance. We have added the HS-SPME-GC-MS spectrograms of three samples in the supplementary materials.We feel sorry for our carelessness.The term here should be changed to "response intensity". The data here involves the total peak area normalization of all peak signal intensities (peak areas) in each sample, i.e., each peak signal intensity is converted into its relative intensity in that spectrum. After normalization, the data is multiplied by 10,000. Redundancy is then removed and peaks are merged, resulting in a value that does not have a unit.
Comments3: What is the effect of using 230C for extraction in the HS-SPME-GC-MS procedure? I think that will affect the compositions severely. Use of such high T for extraction has to be justified.
Response3: Thank you for pointing this out.The reason for setting the injection port temperature at 230 ℃ is related to the following two aspects:
Sample volatility: Headspace mass spectrometry requires the vaporization of the sample so that it can be detected by the mass spectrometer. Some common volatile organic compounds can evaporate at lower temperatures, so the injection port temperature should not be too high at this time; but if the temperature is too low, it may cause some compounds to not fully evaporate, affecting the detection results. After experimental testing, it was determined that 230 ℃ is a suitable temperature range for sample volatilization.
Reduce pollution: Too high or too low injection port temperature may cause problems such as compound decomposition and thermal cracking, thereby affecting the detection results. In addition, if the temperature is too high, it may cause the air around the injection port to be heated and expanded, thereby causing pollution to the sample. Therefore, controlling the injection port temperature within a suitable range can reduce the occurrence of these problems.
Comments4: Any comment on why toxic chemicals such as aromatic benzene/toluene/xylene content is high in the dried mushrooms? Why do vacuum dried mushrooms leach out more aromatic hydrocarbons than the other two?
Response4: Thank you very much for your question. Firstly, vacuum freeze-drying does not lead to the formation of aromatic hydrocarbons in O. raphanipes. The possible reason for this result is that O. raphanipes may have been exposed to an environment containing benzene, toluene, and xylene during its growth or storage process, thereby allowing these compounds to be retained in O. raphanipes. However, heat treatment and infrared radiation may cause the decomposition of aromatic hydrocarbons. Heating can provide sufficient energy for the reaction, causing the chemical bonds of aromatic hydrocarbon molecules to break and then undergo decomposition or other chemical reactions. Infrared radiation is a type of thermal radiation, which can be absorbed by materials and converted into internal energy, and it may also cause the decomposition of aromatic hydrocarbons.
Comments5: As per Table 2, VD samples are lighter in color than the HD and ID samples. Solutions of polysaccharides extracted from these samples seem to show the opposite trend – PS-VD has higher absorbance throughout the UV-vis range indicating the solutions with PS-VD are more colored. Can the authors comment on this?
Response5:Thank you for your question. Regarding the higher absorbance of the polysaccharide (PS-V) dried by vacuum freeze-drying, my explanation is that during the polysaccharide extraction process, the dialysis step can help remove some small molecule impurities, including some sources of color. Dialysis uses the selective permeability of a semi-permeable membrane to separate different sizes of molecules in solution. Small molecules (such as ions, low molecular weight organic molecules, colorants, etc.) can diffuse through the semi-permeable membrane into the dialysis fluid, while large molecules (such as proteins or polysaccharides) are retained within the semi-permeable membrane. Therefore, some of the colorants in the three polysaccharides can be removed by dialysis, reducing experimental errors caused by color.
Comments6: Absorbance values of >2 are error prone as only <1% of the light is transmitted. Authors should report the precision of their absorbance measurements.
Response6: Thank you for pointing out the deficiency, the precision of absorbance measurement is 0.01. This has been added to the text and highlighted.
Comments7: Authors should comment on the choice of drying temperature of 60C. I assume there is an interaction between temperature, time, and drying method.
Response7: Thank you very much for your suggestion. Before conducting the experiment, we carried out a preliminary test to select the best hot air drying temperature among 50°C, 60°C, and 70°C. At the same time, we also referred to the literature "Effect of Drying Temperature on the Volatile Profile and Taste Properties of Flammulina velutipes Root". The final results showed that at 50°C, the overall quality of O. raphanipes was the best, and it was less likely to cause damage to the samples.
We thank the reviewer for finding interest in our manuscript and for the accurate summary of the results of the paper. We tried our best to improve the manuscript and made some changes to the manuscript. Hope that thecorrection will meet with approval. Once again thank you very much for your comments and suggestions.
With kind regards,
Yours sincerely,
Ms. Hou

Reviewer 3 Report
Comments and Suggestions for Authors
Effect of different drying methods on the quality of Oudemansiella raphanipes
The authors investigated the effects of hot air drying, infrared radiation drying and vacuum freeze-drying on the quality of O. raphanipes. They evaluated the rehydration rate, appearance and microstructure quality, volatile components, and physicochemical properties of the polysaccharides in the dehydrated product. They also checked the antioxidant capacity, in addition to other parameters.
The article is well-composed and provides insightful data on the effects of three drying methods on Oudemansiella raphanipes, though the findings are somewhat expected given the extensive research on similar subjects with other biological materials. Vacuum drying (VD) emerged as the superior method across various metrics, aligning with predictions, and this was well articulated in the study's conclusion.
The authors detailed the impacts of drying on aspects like the microstructure, antioxidant properties, and aroma of O. raphanipes. However, it's important to consider the higher cost of vacuum drying. It would be beneficial for the authors to rank the drying methods by their cost-effectiveness, offering a practical guide on when to use each method. While vacuum drying stands out for preserving the quality of O. raphanipes, especially when its shape or size is crucial post-rehydration, there might be scenarios, such as when O. raphanipes is used as an ingredient in soups or sauces, where a less costly drying method could also be appropriate. Identifying the second-best option based on a balance of cost and quality could provide valuable guidance for different applications.
Specific Questions and Clarifications Needed:
1. Table 3 Correction: The abbreviation for vacuum drying should be corrected to "VD" instead of "FD." This ensures consistency across the document.
2. Table 4 Inquiry: There is a noticeable difference in the concentration of acids among the samples. Could you provide an explanation for these variations? Understanding the cause—whether it's due to the drying method, the nature of Oudemansiella raphanipes, or other factors—is crucial.
3. Section 3.6 - Antioxidant Activities Analysis (PS-V, PS-H, and PS-I):
- The abbreviation "Vc" is used without explanation. Please clarify what "Vc" stands for, especially since it's referenced with a scavenging rate of 99.65%.
4. Section 3.7 - Inhibitory Activity Analysis (PS-V, PS-H, and PS-I):
- The term "AC" is mentioned but not defined. Could you elaborate on what "AC" represents? Clarifying this abbreviation is essential for readers to fully grasp the analysis and its implications.
Author Response
Dear Reviewer and Editor:
We would like to thank you for your efforts in reviewing our manuscript titled "Effect of different drying methods on the quality of Oudemansiella raphanipes", and providing many helpful comments and suggestions, which will all prove invaluable in the revision and improvement of our paper, as well as in guiding our research in the future. We have studied your comments point by point, revised the manuscript accordingly. The amendments are highlighted in red in the revised manuscript. The ranking based on the cost-effectiveness of various drying methods has been made and is annotated at the end of the article.
Comments1: Table 3 Correction: The abbreviation for vacuum drying should be corrected to "VD" instead of "FD." This ensures consistency across the document.
Response1: Thank you very much for your question. The FD in Table 3 has been changed to VD and highlighted.
Comments2: Table 4 Inquiry: There is a noticeable difference in the concentration of acids among the samples. Could you provide an explanation for these variations? Understanding the cause—whether it's due to the drying method, the nature of Oudemansiella raphanipes, or other factors—is crucial.
Response2: Thank you for your question. There indeed exists a significant difference in the acid concentration, but we do not have concrete evidence to explain why this phenomenon occurs. However, our guess is that hot air drying, as a heat treatment method, might trigger chemical reactions in O. raphanipes, including the decomposition of amino acids. This might lead to the production of more propanoic acid , 2-methyl-propanedioic acid, butanoic acid . On the other hand, far-infrared drying might cause stress inside O. raphanipes, triggering a series of physiological and biochemical reactions, which might change the metabolic pathways of O. raphanipes and increase the production of 2-methyl-pentanoic acid. Thank you again for your valuable comments. We will continue to think deeply about this question in the future.
Comments3: Section 3.6 - Antioxidant Activities Analysis (PS-V, PS-H, and PS-I):
- The abbreviation "Vc" is used without explanation. Please clarify what "Vc" stands for, especially since it's referenced with a scavenging rate of 99.65%.
Response3: Thank you for pointing out the shortcomings. Here, Vc refers to Vitamin C, which is a powerful antioxidant (in this experiment, its clearance rate for free radicals can reach 99.65%). It can clear free radicals in the body, so it exists as a positive control in this experiment. An explanation of VC has been made in the paper and highlighted.
Comments4: Section 3.7 - Inhibitory Activity Analysis (PS-V, PS-H, and PS-I):
- The term "AC" is mentioned but not defined. Could you elaborate on what "AC" represents? Clarifying this abbreviation is essential for readers to fully grasp the analysis and its implications.
Response4: Thank you for pointing out the issue. In this paper, AC stands for Acarbose, an oral anti-diabetic drug. Its mechanism of action is to inhibit α-amylase and α-glucosidase. These enzymes typically break down polysaccharides and oligosaccharides into monosaccharides in the small intestine, allowing the sugars to be absorbed by the body after digestion. The inhibitory effect of Acarbose can slow down the breakdown and absorption of carbohydrates, thereby reducing the rate of postprandial blood sugar rise. It is present as a positive control in this experiment. An explanation of AC has been provided in the paper and highlighted.
Thanks the reviewer for the helpful comments and appreciation of our work. We tried our best to improve the manuscript and made some changes to the manuscript. We appreciate again for editors and reviewers' warm work earnestly, and hope that the correction will meet with approval.
With kind regards,
Yours sincerely,
Ms. Hou

Round 2
Reviewer 2 Report
Comments and Suggestions for Authors
Authors have clarified most of the questions. Their responses regarding HS-SPME-GC-MS are unsatisfactory.
1. The newly added HS-SPME-GC-MS in SI results have some broad peaks. What are they?
2. Use of 230C for GC-MS analysis I think invalidates the analysis. The authors claim hot-drying @ 60C causes chemical reactions in mushrooms. How would 230C not affect the sample chemistry? I think these results are not valid to be included in the paper.
3. Similarly, the presence of aromatics is probably tied to the use of 230C extraction temperature. If it is coming from the original mushrooms as the authors think, can they show any evidence of this? Authors are claiming that the mushrooms contain toxic chemicals due to environmental exposure. Can they show some positive and negative controls?
Author Response
Dear Reviewer and Editor:
We sincerely appreciate the valuable feedback you have provided. In response to your inquiries, we have supplemented with the relevant attachments. Below, we have addressed each of your questions one by one:
Comments1: The newly added HS-SPME-GC-MS in SI results have some broad peaks. What are they?
Response1: Thank you for your question. Based on the broad peaks you mentioned, we found that they are mainly concentrated before 10 minutes, and we have sorted out the substances corresponding to this time period. The details are uploaded in the attachment 1.
Comments2: Use of 230C for GC-MS analysis I think invalidates the analysis. The authors claim hot-drying @ 60C causes chemical reactions in mushrooms. How would 230C not affect the sample chemistry? I think these results are not valid to be included in the paper.
Response2: Thank you for pointing this out. The GC-MS testing in our paper was conducted by Shanghai Luming Biotechnology Co., Ltd. We have also consulted relevant literature (uploaded in the attachment2,3,4) and confirmed that the temperature is 230°C.
Comments3: Similarly, the presence of aromatics is probably tied to the use of 230C extraction temperature. If it is coming from the original mushrooms as the authors think, can they show any evidence of this? Authors are claiming that the mushrooms contain toxic chemicals due to environmental exposure. Can they show some positive and negative controls?
Response3: Thank you for your comment. To verify the origin of the aromatics in the mushrooms, we conducted GC-MS tests on two identical fresh mushroom samples (the results are in Attachment 5). We found that both samples contained p-Xylene and Benzene,1,3-dimethyl-.
We have strived to contemplate and elucidate the issues. Your professional review plays a crucial role in enhancing the quality of the article. We thank you once more for your support and patience with this piece, and hope that the aforementioned responses will meet your approval.
With kind regards,
Yours sincerely,
Ms. Hou
